# Less Invasive Fixation of Acute Avulsions of the Achilles Tendon: A Technical Note

**DOI:** 10.3390/medicina56120715

**Published:** 2020-12-19

**Authors:** Umile Giuseppe Longo, Vincenzo Candela, Alessandra Berton, Calogero Di Naro, Giovanna Stelitano, Nicola Maffulli, Vincenzo Denaro

**Affiliations:** 1Department of Orthopedic and Trauma Surgery, Campus Bio-Medico University, Trigoria, 00128 Rome, Italy; v.candela@unicampus.it (V.C.); a.berton@unicampus.it (A.B.); c.dinaro@unicampus.it (C.D.N.); g.stelitano@unicampus.it (G.S.); denaro@unicampus.it (V.D.); 2Department of Musculoskeletal Disorders, School of Medicine and Surgery, University of Salerno, 84084 Fisciano, Italy; n.maffulli@qmul.ac.uk; 3San Giovanni di Dio e Ruggi D’Aragona Hospital “Clinica Orthopedica” Department, Hospital of Salerno, 84124 Salerno, Italy; 4Barts and the London School of Medicine and Dentistry, Centre for Sports and Exercise Medicine, Queen Mary University of London, London E1 4DG, UK; 5School of Pharmacy and Bioengineering, Keele University School of Medicine, Stoke on Trent ST5 5BG, UK

**Keywords:** Achilles tendon avulsion, surgical technique, fixation, Achilles tendon technique, sutures

## Abstract

*Purpose:* Nowadays, surgical treatment of acute avulsions of the Achilles tendon represents a hard challenge. There is often the possibility that the calcaneus remains completely uncovered from the tendon, making the reinsertion of its distal stub complex. At the same time, the standard open surgical technique could cause difficult wound healing because of the weak blood supply, the increasing possibility of rupture, and the bacterial contamination. To overcome these risks, less invasive procedures should be considered. *Methods:* We developed an innovative minimally invasive procedure for fixation of acute avulsions of the Achilles tendon employing an integration of four longitudinal stab incisions and one distal semicircular Cincinnati incision. In this way, the distal Achilles tendon stub and the calcaneal insertion are exhibited. *Results:* We basted the tendon through percutaneous sutures performed across the four stab incisions with a Mayo needle threaded with Ultrabraid. The procedure is repeated with another loop of Ultrabraid. After having bruised the calcaneus bone insertion of the tendon, two sites for two suture anchors were prepared using a specific hole preparation device for the anchors’ footprint. Finally, we placed two suture anchors to reinsert the tendon to the calcaneal insertion. *Conclusion:* Our new less invasive technique is a promising alternative optional procedure for the Achilles tendon (AT) avulsion repair allowing clear exposure of the Achilles tendon insertion, maintaining the longitudinal wholeness of the dermis, and minimizing possible associated complications.

## 1. Introduction

An acute avulsion of the Achilles tendon (AT) from the calcaneus is not as frequent as the rupture of the AT, usually located 2–6 cm proximally to the calcaneal insertion [1,2,3,4]. This kind of avulsion, which usually occurs near the distal border of the tendon, requires surgical trials due to the deficiency or, more frequently, the total lack of tendinous tissue at the level of the calcaneal insertion of the AT. These conditions make the reinsertion of the distal stump of the AT difficult [5]. Bibbo et al. described a procedure a with transcalcaneal suture for the reparation of AT avulsions, by an incision full and lengthwise [6].

Bibbo’s method grants calcaneal reinsertion of the Achilles tendon only in patients with a minimal tendinous defect. If the tendon tissue shows a significant defect, the surgeon should prefer the use of a tendon graft to remove the gap and recreate a stable structure. The standard open surgical technique for the AT rupture repair could provoke difficult healing of the surgical wound because of the weak blood supply of the involved tendon area and higher possibility of bacterial infection and tendon rupture. The extensive exposition of the open surgical technique can cause a substantial alteration both of subcutaneous tissues and the paratenon together with a high risk of development of peritendinous adhesions. Complications after surgical repair can be classified into three different categories: minor, general, and major [7]. The first category refers to wound complications. The second one includes clinical symptoms such as discomfort, swelling, and weakness. The third and last category describes the true problem of this technique. It covers the development of deep or chronic infection, tendon lengthening, pulmonary embolism, deep vein thrombosis, and death. Achilles tendon re-rupture and sural nerve damage must be considered as additional severe complications [8]. As described above, open surgical techniques have shown an increased risk of failure of wound healing due to insubstantial blood supply and the subsequent risk of bacterial contamination (9%). To overcome these risks, less invasive procedures have been described for the treatment of AT tendinopathy [9,10,11], acute AT ruptures [12], chronic AT ruptures [13,14,15,16,17,18], chronic AT avulsions [3]. As far as we know, no less invasive fixation has been mentioned for the surgical treatment of acute avulsions of the AT. At the same way, the standard technique approved for the treatment of tendon avulsions in the absence of bone pieces or in the presence of little fragments in which, however, fixation to the calcaneus bone becomes impossible does not exist. We described a new minimally invasive surgical procedure for fixation of acute avulsions of the AT using an integration of four longitudinal stab incisions and one distal Cincinnati cut, showing the distal AT piece and its insertion. We used two suture anchors to reinsert the tendon to the calcaneal insertion [5,19]. Through the use of this procedure, broad exposure to the insertion of the AT is achievable. Percutaneous suturing of the AT provides careful preparation of the tendon end reducing skin cut, hence protecting against wound rupture. The less invasive approach technique has many advantages, including the accurate reinsertion of the AT with minimized chances to damage the sural nerve, quicker recovery period, shorter hospitalization, and enhanced functional outcomes in comparison to the conventional invasive methods [19].

## 2. Materials and Methods

We have refined a new less invasive procedure for fixation of acute avulsions of the AT utilizing an integration of the percutaneous suture method and the mini-open access. The case reported in our study is that of a man of 47 years of age affected by an AT avulsion and undergone the present procedure in May 2018 at the University Campus Bio-Medico in Rome. The surgical procedure has been performed by one of the authors (U.G.L.). The diagnosis of AT avulsion was made on the basis of clinical presentations and the physical examination, ankle Χ-ray, and MRI. The patient’s mechanism of injury was classified as atraumatic (AT avulsion occurred during a sportive activity). The patient was not affected by other Achilles tendon disorders, neither he presented with a history of corticosteroid AT injections. The elapsed time from injury to surgery was 9 days. The patient’s body mass index (BMI) was 26.5 kg/m^2^. The patient read and signed the written informed consent form. Ethics approval was deemed unnecessary according to the Italian legislation.

### 2.1. Clinical Evaluation

The patient’s disease was evaluated through the use of subjective outcomes, such as the Victorian Institute of Sports Assessment–Achilles (VISA-A) score [20], the Achilles Tendon Rupture Score (ATRS) [21], and time to return to activities. The results were assessed by one author five times: before the surgery; 3, 6, 12, and 24 months after the operation. The examiner evaluated persisting symptoms of heel pain and stiffness, the patient’s capacity to perform the heel rise test, the Thompson test, and wound characteristics. In the end, to verify whether complete healing of the Achilles tendon was achieved, an MRI was performed one year postoperatively.

### 2.2. Technical Description

The procedure was performed under locoregional anesthesia with the patient in the prone position. A calf tourniquet was applied at the root of the thigh. The skin was prepared regularly. Pre-surgical anatomical signs such as the tangible tendon defect and the calcaneal tuberosity were identified. Five skin cuts were performed. Four longitudinal cuts at the medial and lateral margin of the tendon around 5 cm proximally to the tendon gap were made. The last incision was a 5 cm semicircular Cincinnati incision over the area of AT insertion [19] (Figure 1). Distal AT pieces were mobilized and freed of all peritendinous adherences. After finding that the avulsed AT could reach the calcaneus, a 5 cm Mayo needle was filleted with a loop of Ultrabraid (white UHMW polyethylene cobraided with blue monofilament polypropylene; The Smith & Nephew, York Science Park, Heslington, York YO10 5DF, UK) and it was rubbed transversely through two proximal cuts inside the tendon fibers (Figure 2). Then, all tendon extremities were rubbed distally from just proximally to the transverse Ultrabraid passage between tendon fibers to move out of the cornerwise stab cut. A successive diagonal pass was carried out to the transverse incision across the broken tendon. Finally, the safety of the suture has been verified through the pull with both ends of the Ultrabraid distally. Subsequently, an additional double loop of Ultrabraidwas put through using a similar technique. After having bruised the calcaneus bone insertion of the tendon, a specific hole preparation device for anchor footprint was used (Figure 3). Two Polyether ether ketone (PEEK) anchors (FOOTPRINT Ultra and TWINFIX◊ Ultra Suture Anchor, Invibio^®^ polymer; Endoscopy, Smith & Nephew, Inc., Andover, MA 01810, USA) of 5 mm were positioned parallelly with an inclination of approximately 45° in relation to the plantar surface (Figure 4).

One double loop of Ultrabraid was passed inside the guidewire of the medial anchor. The same procedure was also applied for the lateral anchor with the other double loop of Ultrabraid. After unhooking the retention suture from the inserter handle and pulling to lock the suture by the handle, the retention suture was discarded before applying tension. The same passages were performed for the other anchor. After the application of the right tension, the sutures were blocked simultaneously. The tendon was fixed, therefore, to the calcaneal insertion with suture anchors, realizing a large tendon–bone interface. For the final suture, a Vycril Rapide zero–four wire (Polyglactin 910 braided absorbable suture; Johnson & Johnson, Brussels, Belgium) was employed. Lastly, a strip was utilized and a bivalve removable scotch cast in complete plantar flexion was positioned. Full weight-bearing was provided the day immediately post-surgery, with the cast in complete plantar flexion.

## 3. Results

The patient claimed the removal of all the pre-surgery symptoms. The VISA-A score increased from a preoperative score of 26 to postoperative scores of 75, 79, 82 and 91 at 3, 6, 12, and 24 months, respectively. Likewise, the Achilles Tendon Total Rupture Score (ARTS) score improved from a pre-surgery score of 23 to post-surgery scores of 69, 80, 82, 90, 95, respectively. The patient resumed their everyday, work, and sportive activities 5 months, 3 months, and 12 months after the surgery (Figure 5). The patient could perform heel rise 14 weeks after the surgery. The Thompson sign was always negative after the surgery. Wound healing was completed without signs of retraction or infection. MRI examination at 12 months was negative for AT damages.

## 4. Discussion

Even today, the worldwide recognized standard treatment for AT avulsions does not exist. The literature provides only limited information concerning the treatment choice for this type of damage, the lack being without any doubt related to the rarity of such lesions. One of the main problems of reinsertion surgery of an avulsed AT is skin closure, because close attention should be paid to the gastrocnemius muscle and skin retraction. In the current literature, the great debate about the exact point of the surgical incision remains unsolved because of the possible complications, such as the dehiscence of the surgical wound, infections, adhesions, and nerve damage [5]. The majority of surgical techniques use longitudinal skin incisions of variable lengths, even though several transverse incisions have been reported. The surgical technique described in our study uses four longitudinal stab skin incisions executed parallelly on both the lateral and the medial side of the tendon. These kinds of incisions allow the introduction of the Mayo needle with the double loop of Ultrabraid to perform suturing of the tendon. At the same time, such small incisions reduce the risks of dehiscence of surgical wounds and post-operative infections. The consequent problem of a reduced surgical visual is overcome thanks to concomitant use of a transversal incision of about 5 cm performed at the level of the injury. This additional incision improves the view of the field of action and allows stretching the distal stump of the tendon. The semicircular Cincinnati incision above the site of the avulsion is made as small as possible to create the proper visual window to perform correct cleaning of the tendon and optimally prepare the site of insertion on the calcaneus bone. It is important to emphasize that our surgical procedure reduces the risk of potential nerve damage related to the standard open surgery. Considering that the sural nerve meets the lateral edge of the AT, the execution of small longitudinal skin cuts made parallelly to the nerve makes nerve damage improbable [19]. Furthermore, should nerve damage occur, it is more likely to involve neuropraxia rather than transverse axonotmesis or neurotmesis [19]. Furthermore, this wound model could provide nerve renewal on its cover and reduce the formation of a neuroma. The choice of employing suture anchors is justified by their ability to allow correct tendon retention and provide secure reattachment of a tendon to the bone, with a pull-out strength superior in comparison to transosseous sutures. After the surgery, a bivalve removable scotch cast is applied to keep the ankle at rest, preventing rupture of the reinsertion and supporting its consolidation. It has been widely reported in the literature that soft tissue vascularity is maximal at twenty degrees of plantarflexion, while at forty degrees, the skin’s hematic flow is reduced by 49%. For this reason, performing the repair can affect wound healing [22]. This post-operative modality has been extensively applied for both acute and chronic AT ruptures [10,11,13,23]. To our knowledge, this is the first less invasive surgical procedure performed for the treatment of acute avulsions of the AT. Among the principal advantages of this technique the ease of performance and reproducibility must be considered. Another essential advantage consists in the possibility to not use expensive materials. However, we are aware that the use of this technique is limited only by the cases in which, by pulling the distal stump of the tendon, it is possible to bridge the gap to be able to reinsert. Otherwise, it is necessary to practice other techniques, such as augmentation with allografts to fill the gap and allow adequate reinsertion. Our technique is based on the combined use of a modified Bunnell suture for the AT, but in a percutaneous way, trying to guarantee safer wound healing together with less skin irritation. Requiring a small incision, predictably, this technique results in smaller scar tissue in the surgical area, which could bring great benefits to clinical results.

## 5. Conclusions

Our new less invasive technique is a promising alternative optional procedure for the AT avulsion repair, allowing a clear exposure of the Achilles tendon insertion, maintaining the longitudinal wholeness of the dermis, and minimizing possible associated complications. The indications for this type of surgical treatment could be particularly helpful for the subjects predisposed to wound complications, such as those affected by vascular diseases and diabetes. Undoubtedly, more cases with longer follow-up are required to define the real advantages of this technique.

## Figures and Tables

**Figure 1 medicina-56-00715-f001:**
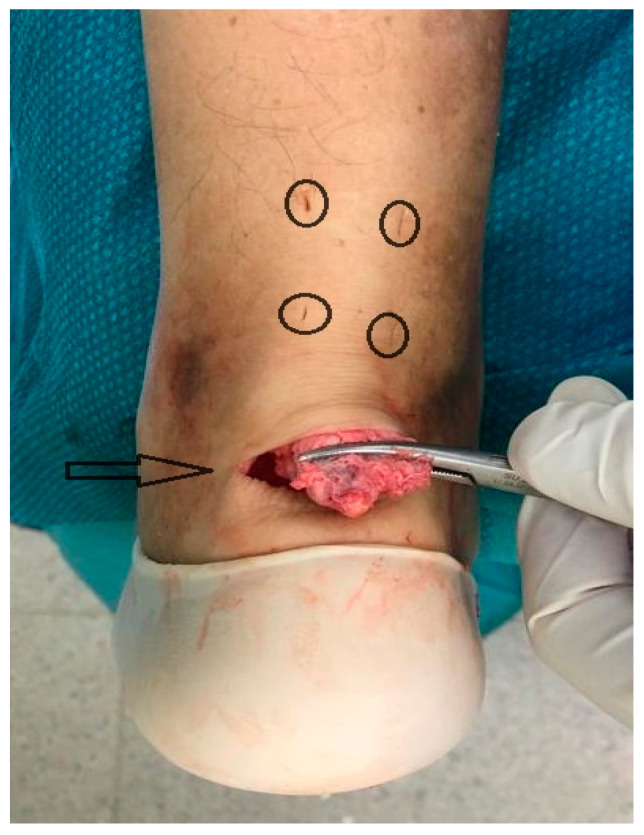
Four longitudinal stab incisions are made laterally and medially to the tendon (O), and a semicircular Cincinnati skin incision is made over the area of achilles tendon (AT) insertion (->).

**Figure 2 medicina-56-00715-f002:**
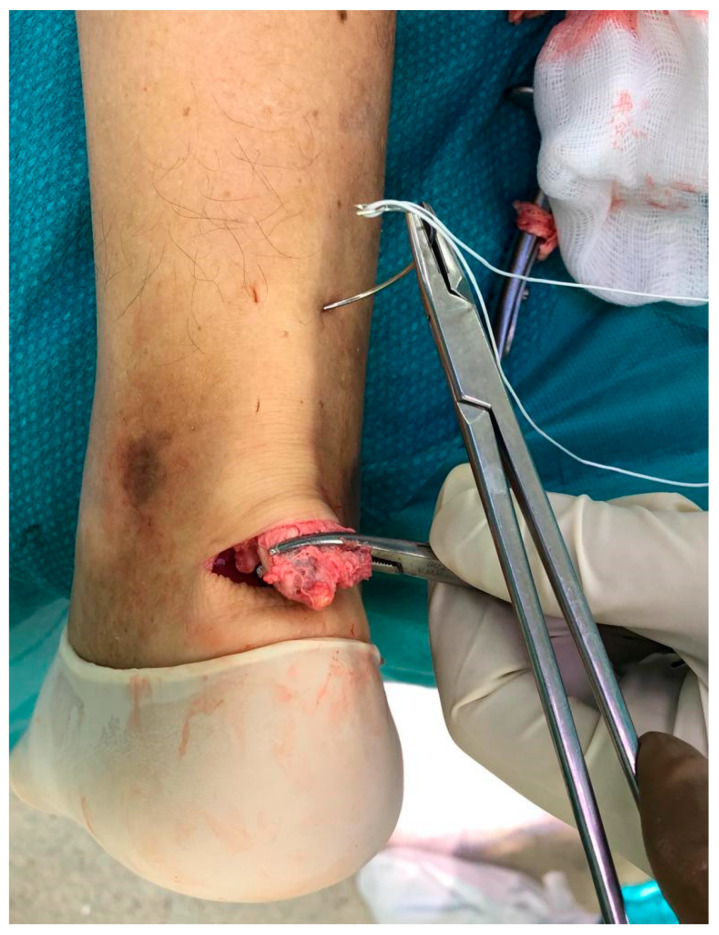
A loop of Ultrabraid is passed transversely between the proximal tendon bulk using a Mayo needle.

**Figure 3 medicina-56-00715-f003:**
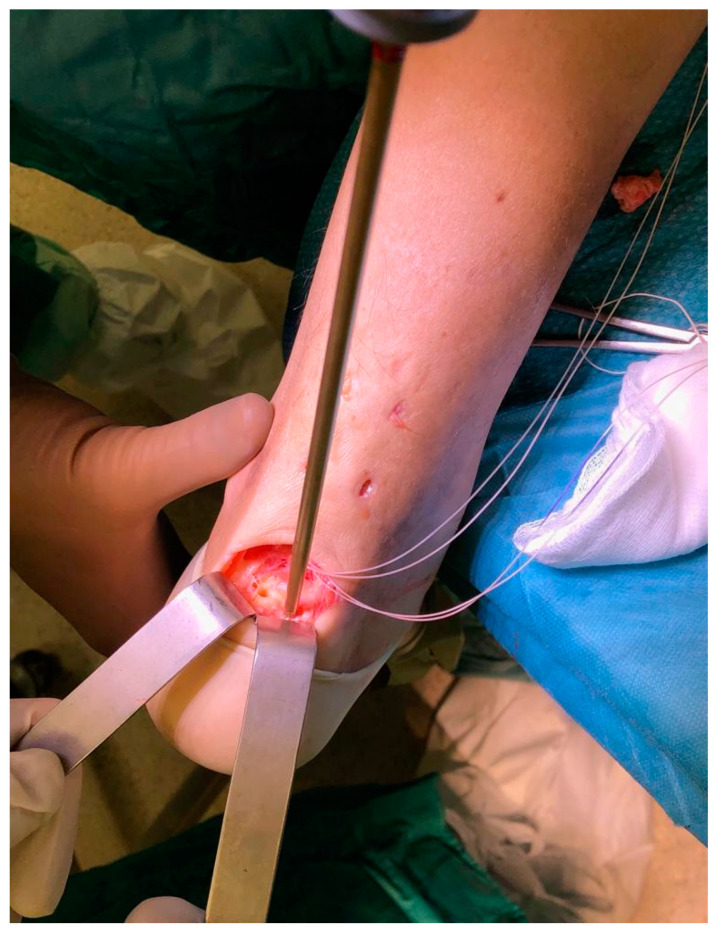
Two insertion sites are prepared using a specific hole preparation device for suture anchors.

**Figure 4 medicina-56-00715-f004:**
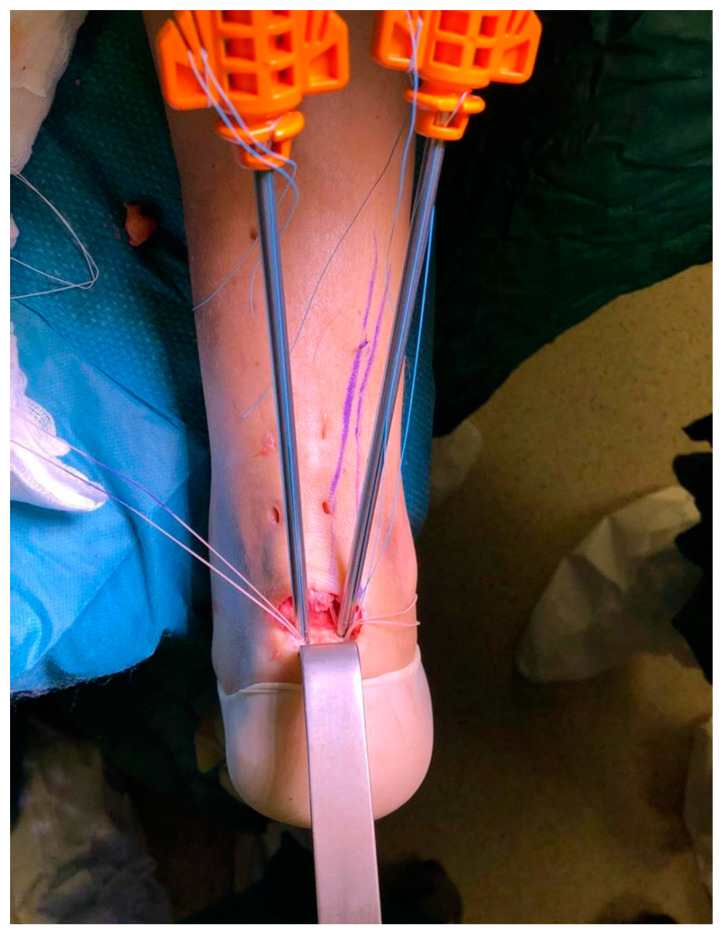
Two suture anchors with their cannulas inserted in the calcaneus bone.

**Figure 5 medicina-56-00715-f005:**
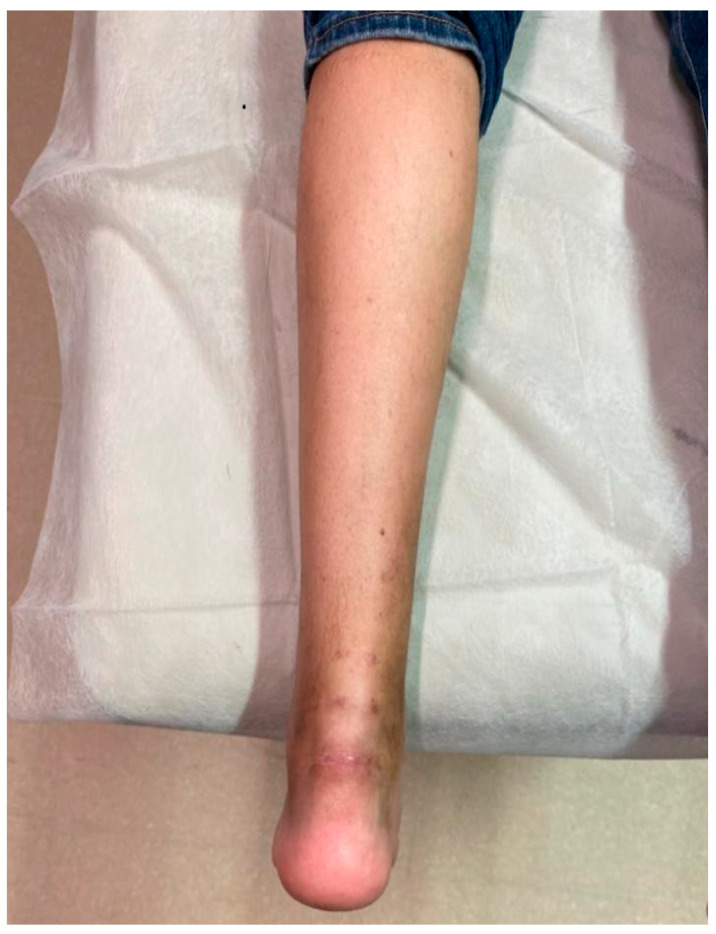
Follow-up at three months.

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
