# Peer review of "Less Invasive Fixation of Acute Avulsions of the Achilles Tendon: A Technical Note"

_medicina, 2020, doi:10.3390/medicina56120715_

Round 1

Reviewer 1 Report

Less Invasive Fixation of Acute Avulsions of The Achilles Tendon

The paper is acceptable for publication as a technique only. The authors did not state how many patients had this procedure, the results, how long the authors have been using this procedure, the advantage vs disadvantage of surgery or when the procedure is recommended, and for which patient groups. The authors did not include an average recommended follow-up, if any technical difficulties were encountered, or if the patients experienced any nerve damage, wound or infection. There was no mention of the strength of the repair or strength of the Achilles tendon after the repair.

Although the procedure provides an alternative technique in the reconstruction of an Achilles tendon rupture, the technique did not provide details that the surgeon will need, especially when sharing information with their patients. The literature does not provide the surgeon with any valuable information that they can provide to the patient when explaining why they chose the procedure.

Author Response

Dear Reviewer,

Thank you for your suggestions.

We have tried to add the missing information as required.

This procedure has been performed for the first time in the patient case described in the present article (line 83). The advantage and disadvantage of this technique are described in the discussion section (lines 172- 176; 187- 190; 202- 203; 204- 207).

We have also specified the choice group of patients eligible for this kind of surgery (line 215- 217). 

Surgical technique description has been improved inside the material and method section. The patient follow-up has been evaluated up to 24 months but, as specified in the conclusion, longer follow-up is needed. Patient’s results have been reported on a separated section into material and method.

The choice of this procedure is due to its simplicity of execution, reduced postoperative complications as widely explained throughout the article.

Reviewer 2 Report

The authors describe a novel mini-invasive surgical technique for the re-attachment of Achilles tendon avulsion ruptures back to calcaneal bone. The manuscript is a description of a new surgical technique and should be classified as a case report as far as I understand it.

I have following criticism for the manuscript:

  1. The presentation in English is very poor. The manuscript needs to be re-written/edited by someone who has proper English skills.
  2. No results are presented. The authors should describe how many patients have been operated with the technique and whether there have been complications as the aim of the novel technique is to reduce the number of complications.
  3. As there is no comparison to another treatment, its should be clearly stated that the potential benefit of the novel technique (in avoiding the complications) cannot be properly assessed in the Achilles tendon avulsion ruptures.
  4. The reference list contains large number of references on Achilles tendinopathy, a chronic overuse disease of the Achilles tendon. Please remove all unnecessary self-citations that are not related to Achilles tendon avulsion ruptures.

Author Response

Dear Reviewer,

As required, we provide to change our article in the case report form.

We have tried to improve the English form, rewriting entirely the paper.

Results have been reported in a separated section, as well as the other missing information has been added inside the material and method section.

The difference among out technique and the other standard surgical procedure describes in literature have been cited in terms of advantage and disadvantage in the discussion section.

We provide to remove the unnecessary citations.

Round 2

Reviewer 1 Report

The authors made corrections according to the reviews recommendations. This paper is acceptable in its current format